# Improved Perceptual Quality of Traffic Signs and Lights for the Teleoperation of Autonomous Vehicle Remote Driving via Multi-Category Region of Interest Video Compression

**DOI:** 10.3390/e27070674

**Published:** 2025-06-24

**Authors:** Itai Dror, Ofer Hadar

**Affiliations:** School of Electrical and Computer Engineering, Ben-Gurion University of the Negev, Beer-Sheva 8410501, Israel

**Keywords:** self-driving cars, teleoperation, video encoding, HEVC, ROI compression, semantic segmentation

## Abstract

Autonomous vehicles are a promising solution to traffic congestion, air pollution, accidents, wasted time, and resources. However, remote driver intervention may be necessary in extreme situations to ensure safe roadside parking or complete remote takeover. In these cases, high-quality real-time video streaming is crucial for remote driving. In a preliminary study, we presented a region of interest (ROI) High-Efficiency Video Coding (HEVC) method where the image was segmented into two categories: ROI and background. This involved allocating more bandwidth to the ROI, which yielded an improvement in the visibility of classes essential for driving while transmitting the background at a lower quality. However, migrating the bandwidth to the large ROI portion of the image did not substantially improve the quality of traffic signs and lights. This study proposes a method that categorizes ROIs into three tiers: background, weak ROI, and strong ROI. To evaluate this approach, we utilized a photo-realistic driving scenario database created with the Cognata self-driving car simulation platform. We used semantic segmentation to categorize the compression quality of a Coding Tree Unit (CTU) according to its pixel classes. A background CTU contains only sky, trees, vegetation, or building classes. Essentials for remote driving include classes such as pedestrians, road marks, and cars. Difficult-to-recognize classes, such as traffic signs (especially textual ones) and traffic lights, are categorized as a strong ROI. We applied thresholds to determine whether the number of pixels in a CTU of a particular category was sufficient to classify it as a strong or weak ROI and then allocated bandwidth accordingly. Our results demonstrate that this multi-category ROI compression method significantly enhances the perceptual quality of traffic signs (especially textual ones) and traffic lights by up to 5.5 dB compared to a simpler two-category (background/foreground) partition. This improvement in critical areas is achieved by reducing the fidelity of less critical background elements, while the visual quality of other essential driving-related classes (weak ROI) is at least maintained.

## 1. Introduction

As the automotive industry progresses toward full automation (SAE level 5), a practical hybrid model combining autonomous driving with human teleoperation is emerging as a critical solution [1]. This approach allows a remote human operator to intervene and safely control a vehicle when its onboard automation encounters complex scenarios it cannot resolve, such as unexpected road work, system malfunctions, or dense traffic [2]. The viability of this model, however, depends on providing the remote driver with a high-quality, real-time video feed over variable and demanding cellular networks [3].

For teleoperation to be a safe and scalable solution, strict performance requirements must be met. The total human–machine loop, often measured as the Glass-to-Glass (G2G) delay, must be minimized. Experiments over 5G have reported G2G delays of around 202 ms, which was deemed safe only for low speeds [4], while other studies suggest that a target of under 100 ms is crucial for a driver to respond effectively [5]. Meeting this low-latency requirement necessitates aggressive video compression, but this creates a critical trade-off with image quality.

This trade-off becomes particularly dangerous when considering how a remote operator perceives the scene. Standard video compression can degrade the legibility of small but vital objects. Even conventional region of interest (ROI) video compression, which partitions a frame into a simple foreground and background, often fails in this context. Critical objects like traffic signs and lights are frequently overshadowed by larger foreground elements like the road or other vehicles, preventing them from receiving sufficient bandwidth to ensure they are clearly visible. This perceptual ambiguity introduces an unacceptable safety risk. To address this specific challenge, this paper introduces an enhanced multi-category region of interest (ROI) compression framework. Our approach moves beyond a binary ROI partition by using semantic segmentation to classify the scene into three distinct tiers based on driving importance:Strong ROI: Containing the most critical objects for driver decision-making, specifically traffic signs and traffic lights.Weak ROI: Containing other essential contextual elements, such as the road, lane markings, pedestrians, and other vehicles.Background: Containing non-essential elements like sky, trees, and buildings.

By dynamically allocating bandwidth according to this three-tiered classification, our method significantly improves the perceptual quality of the most critical objects without degrading the surrounding context. This study makes the following contributions:We design and implement a multi-category ROI compression system that prioritizes traffic signs and lights to improve safety in remote driving teleoperation.Using a photo-realistic driving simulator, we demonstrate that our method improves the perceptual quality of traffic signs and lights by up to 5.5 dB compared to a conventional two-category ROI approach.We validate that this improvement is achieved while maintaining the visual quality of other essential driving elements in the weak ROI, ensuring a comprehensive and clear view for the remote operator.

## 2. Background

Developing a fully automated self-driving car beyond the Society of Automotive Engineers (SAE) level 3 is challenging [1]. Designing a fully self-maneuverable vehicle that can drive safely under any driving conditions, including fog, heavy rain, road obstacles, complex roadside works, impolite and incapable drivers, and pedestrians on the road is not easy. This problem is much more complicated than initially thought by many of the designers.

A fallback solution allows for the remote driver control of the vehicle when the self-driving car’s automation encounters difficulty in handling the driving tasks independently. Most of the time, driving is autonomous, and only in rare cases, such as extreme weather, very dense driving situations, and obstacles, will the remote driver take control of the vehicle. To make such a system cost-effective, in most remote driver interventions, the distant driver will drive the car quickly to a safe and nearby location, and then the support will arrive at the vehicle.

Reliable and safe teleoperation by a remote driver in a distant control room requires high-quality, low-delay, and reliable video communication. The available cellular 4G and 5G network bandwidth varies significantly when a car moves and switches from one cell to another. Furthermore, varying distances between the vehicle and the network antennas and network congestion make a video broadcast over the cellular network demanding [3].

The required Glass-to-Glass (G2G) delay for the video transmission is measured as the time passed from the rays of an event hitting the camera’s lens until the event is displayed on the teleoperator’s monitor [4,6,7]. The authors of [4], in an experimental evaluation of a Kia Soul EV with a teleoperated driving interface over 5G, reported an average G2G of 202 ms and RTT (Round Trip Latency) of 47 ms. They concluded that their teleoperation setup is safe at low speeds of less than 20 km/h.

A study of the influence of the delay in a multiparty racing car game shows that a player can respond well if the delay is smaller than 100 ms. Therefore, reliable low-delay video transmission by a teleoperator field is crucial for safe driving [8].

This study proposes an improvement in video quality by applying multi-category ROI compression. Instead of parsing the image to the ROI and background, we rank the ROI according to their importance for remote driving. We used two categories: weak and strong ROI. Critical objects for driving are traffic signs and lights, which are assigned much of the relative (to their size) bandwidth, enhancing their visibility while at least maintaining the quality of the weak ROI and migrating bandwidth from the background.

The video encoder compresses video coding blocks to a higher quality if they contain more critical elements for safe driving. Conversely, coding blocks that are less important for safe driving are considered as the background and are compressed with a lower quality.

The requirement for a fast teleoperator response and low-delay video causes the granularity of the rate control (RC) to be at the level of a single frame. However, many video applications are less prone to real-time performance and have RC granularity at the Group of Picture (GOP) level or time frame (e.g., a second).

The car’s edge computer analyzes the video and detects which parts are more important for remote driving and which are less critical. Several image parsing and comprehension methods are available. First, deep learning (DL) semantic segmentation tools [9] can classify every image pixel by its class, and then it is possible to assign an image block to an ROI category based on the block content. Second, DL object detection methods [9] can be trained to find the bounding boxes of essential objects such as vehicles, traffic lights, and traffic signs. The coding blocks are mapped and assigned ROI priorities only if they contain parts of the crucial objects. Third, the car’s Lidar (light detection and ranging) depth maps [10] can distinguish between ROI and non-ROI coding blocks according to their distances from the vehicle. After mapping the image into non-ROI and ROI coding blocks, the encoder allocates bandwidths to both regions. Finally, it assigns the bitrates according to the temporal bandwidth available from the network operator, the relative sizes of the ROI and non-ROI, and the predefined weight that defines the normalized bitrate ratio of the non-ROI to ROI.

This study proposes further improving video quality by applying multi-category ROI compression. Instead of parsing the image into the ROI and background, we used two categories: weak and strong ROI. Critical objects for driving, that is, traffic signs and lights, are assigned much of the relative (to their size) bandwidth, enhancing their visibility while at least maintaining the quality of all other objects that are also important for the distant driver.

This work relies on several concurrent technologies: self-driving cars, the teleoperation of autonomous vehicles when necessary, complementary metal-oxide-semiconductor (CMOS) video cameras, Lidar sensors, radars, a Global Positioning System (GPS), High-Efficiency Video Coding (HEVC), also known as the H.265 video compression standard, a Cognata [11] autonomous vehicle photo-realistic simulator, and deep neural networks that can analyze the video image for the following tasks, including object classification, object recognition, object tracking, semantic segmentation, instance segmentation, etc.

### 2.1. Self-Driving Cars

Self-driving cars will change our public or private attitude toward transportation, driving, car ownership, car safety, car accidents, and loss of working hours.

The SAE defines five levels of self-driving car automation [12]. Level 0 represents a conventional car in which the human driver controls the vehicle. Most of the cars available today are at level 0. However, human driver functionalities migrate into the car’s automation as the SAE level increases. At level 1, the vehicle has driving assistance such as automatic braking, lane assistance, and adaptive cruise control. The car’s automation can control its speed or the steering, but not both. The driver is responsible for monitoring the road and must immediately take over if the assistance system fails.

Level 2 refers to partial automation. The car can control steering, acceleration, and braking if possible. The automation is responsible for more complicated tasks such as changing lanes, responding to traffic signals, and looking for hazardous conditions. The driver’s hands should immediately take control of the vehicle when requested. Examples of level 2 systems are Tesla Autopilot and Mercedes-Benz Driver Assistance Systems. From level 3 and above, car automation can take over in certain traffic conditions, such as assistance in traffic jams or acting as a highway chauffeur. In level 3, the driver can perform tasks other than driving. Texting on a cell phone, reading, writing, or relaxing is possible. The driver should be prepared to control the vehicle if the automation cannot handle complex driving situations or if there is a malfunction in the car’s automation.

Level 4 indicates that the car is fully automated. It can handle all driving tasks, even under challenging driving conditions. At level 4, there is no need for human driver intervention. The driver can take control and steer the car by hand in case of a system malfunction or personal desire for self-driving; however, the vehicle generally acts as a self-driving device. Currently, the only level 4 self-driving car operating is the Waymo fleet. The Waymo case is still a level 4 in a narrow sense because it is available only in limited locations. The system is not entirely autonomous and does not function in bad weather, and remote human assistance can guide the automated vehicles under certain conditions [2].

Finally, level 5 represents an autonomous vehicle without human interaction. At this level, cars are autonomous robots that transport passengers and goods independently.

### 2.2. Teleoperation of Almost Self-Driving Cars

The highest level of automation available for public sale is level 3, and even its availability is minimal [13]. It is limited only to a few mapped highways, the speed is limited to 40 mph, and it is allowed only during the day and in good weather conditions. Migration from a level 3 car to a fully automated level 4 vehicle or a completely autonomous vehicle at level 5 is challenging. Teleoperation can fill the technological gap between an autonomous vehicle, which is imperfect and requires human assistance on rare occasions, and the desired fully robotic system. The remote driver controls the computerized vehicle in case the car’s automation malfunctions, or an unexpected driving situation arises that the vehicle cannot handle safely. The idea is to allow automation to control the vehicle fleet at 99.9%, while the remote drivers will intervene in the remaining 0.1% of the driving hours [14]. The teleoperation of the car should receive a high-quality video with minimal delay between the camera and the remote driver’s monitor. The round-trip signal delay comprises the car’s camera-to-remote monitor delay, the teleoperator response time, and the time required to transmit the steering wheel and pedal commands from the distant driver to the autonomous car. This human–machine closed-loop time should be at most 170 ms [15], leaving much less time for signal travel between the camera and the teleoperator’s monitor.

### 2.3. Use of 4G and 5G Cellular Networks for Teleoperation

Experiments with teleoperation over 4G networks [3,16] showed that the upload and download requirements are adequate under many conditions. However, the network must be more robust to enable smooth teleoperation. These experiments with the 4G network suggest that the 5G network may be acceptable for reliable teleoperation. A remote driver should have a continuous HD video of high quality and a maximum delay of the image between the camera sensor and the operator monitor of no more than 100 ms. This short interval also contains the video encoding, decoding, and 5G packet transmission time. Therefore, successful remote teleoperation requires high-quality video transmission with a minimum delay over a 5G network. This task is very challenging because the network quality is not uniform, and there are many driving conditions under which the network bandwidth degrades significantly.

### 2.4. The Cognata Driving Simulator

Driving scenarios were generated using the Cognata driving simulator. Cognata produces driving movies for autonomous vehicles through controlled simulations [17]. In addition, the Cognata simulator can create photo-realistic videos that are close to reality. Photo-realistic driving scenarios are essential for video compression research because their compression results are close to the compression of real-life scenarios rich in fine details. However, non-photo-realistic simulators yield simplistic scenes with low spatial complexity, leading to much smaller compressed files than actual and natural driving scenes. The user can decide where to place cameras, Lidars, radars, GPS, and lane detectors on the car’s layout. After performing a driving simulation according to a modified driving scenario, the user can download the response of each sensor provided by a contiguous video or a collection of non-compressed images in Portable Network Graphics (PNG) format. Figure 1a shows a snapshot of a driving scenario from the car’s front camera. Figure 1b shows the ground truth semantic segmentation map of Figure 1a according to the pixel classes of the Cognata simulator. The semantic segmentation sensor has around 50-pixel classes, where every class has its unique color. For example, the sky pixels are colored blue, and the road pixels are colored gray. The Cognata pixel classes are based on earlier works. A key example of such influential earlier work is the Cityscapes dataset [18], which is notable for scene understanding and semantic segmentation for self-driving cars. It comprises an extensive collection of urban street scenes captured from different cities, with pixel-level annotations that label various objects and areas within the images. The dataset is frequently used to train and evaluate algorithms designed to understand and interpret the visual information present in urban environments. Cityscape has 30-pixel classes.

### 2.5. Concurrent Video Coder Encoders (Codecs)

A crucial component of a teleoperation system is a video encoder/decoder. Therefore, it is essential to select an appropriate video compression standard. An aging standard may be robust and rich in practical software and hardware implementations, but it yields poor rate-distortion (RD) performance. However, a very recent standard may not be mature yet and lack practical and available real-time deployments.

In this research, we chose HEVC (H.265) [19] over Advanced Video Coding (AVC, H.264) [20] because of its superior advantage in terms of bitrate savings. The results in [21] show the clear benefits of HEVC over AVC and Google’s open-source VP9 [22]. The comparative study in [23] compares H.265, H.264, and VP9 for low-delay real-time video conferencing applications. The results show that H.265 is better than H.264 and VP9 by 42% and 32.5%, respectively, regarding bitrate and for the same peak signal-to-noise ratio (PSNR).

In addition, contemporary hardware can manage the higher coding complexity of HEVC. Although there are new compression standards, such as the Versatile Video Coding (VVC)/H.266 [24], AV1 [25], and MPEG5 [26] their real-time practical encoding implementations are yet to be made available. According to Bitmovin’s annual Video Developers Report [27], the AVC is still the most dominant video codec in current video applications. HEVC is the second. However, if we look at the Codecs that developers plan to use in future designs, we can see that HEVC excels over AVC.

The H.265 encoder partitions an image into Coding Tree Units (CTUs). Its size can be as high as 64 × 64 pixels [19]. The encoder can recursively subdivide each CTU into Coding Units (CUs), ranging from 32 × 32 to 8 × 8 pixels. Each Quantization Group (QG) has a CU size according to the CTU partitions and can be assigned a separate QP [19]. Varying the QP controls the compression quality. The standard defines and allows for fine control such that each CU has a separate QP. However, for simplicity, in this study, we limited the compression quality granularity to the size of the entire CTU.

### 2.6. Rate Control (RC)

The RC algorithm is a lossy compression optimization method that fits the compressed video size into a bandwidth constraint. The target is to minimize the video distortion (*D*) at a given encoder rate for reconstructed images from the lossy compressed representation. The RC algorithm fits the compressed video size into the bandwidth limitations of the network. One of the most common measures of *D* is the mean square error (MSE). There is always an RD tradeoff. When the network allows a higher *R*, it achieves smaller *D* values. RC methods have changed significantly during the long journey to attain modern video compression since the 80s. With the development of new video encoding standards, rate-control methods have also been adjusted to meet recent changes and requirements.

In [28], the refresh rate, QP, and movement detection thresholds were adjusted to regulate the very coarse bitrate of the early H.261 standard. The authors of [29] proposed a quadratic relationship between QP and *R* for the MPEG2 and MPEG4 compression standards:(1)$$R=\frac{a}{QP}+\frac{a}{QP^{2}}$$
where a and b are calculated based on the encoding of previous frames.

The authors of [30] showed that the *R-λ* RC methods proposed for HEVC outperformed the previous HEVC RC, which is based on *R-Q* model dependencies and is part of the HM reference software (the R-λ model became a prominent part of the rate control mechanisms in HM around HM 10.0 and later versions).

Lagrange multipliers help us select an optimal point along the RD monotonically decreasing convex function [31]. It uses the Lagrange multiplier optimization, where λ, the Lagrange multiplier, depends on the frame complexity. The granularity of this algorithm can be an entire GOP, a unit of time, or at the frame or CTU levels. It can assign a separate QP to each CTU. The Joint Collaborative Team on Video Coding (JCT-VC) included this method in its HEVC Test Model. The *R-λ* method is also implemented in Kvazaar [32] as the default RC method, and we will review it.

The objective of RC is to minimize the overall *D* at a target bitrate, *R*.(2)$$\underset{{\left\{r_{i}\right\}}_{i=1}^{M}}{\min}D=\sum_{i=1}^{M}d_{i}\;\;\;\;\;s.t.\sum_{i=1}^{M}r_{i}\leq R$$
where $d_{i}$ and $r_{i}$ are the distortion and target bits, respectively, for the *i*’th CTU. *M* is the total number of CTUs in a frame. The Lagrange method converts the constrained Equation (2) into an unconstrained optimization equation. Equation (3) can be solved by taking the derivatives with respect to $r_{i}$ and comparing the values to zero. Li et al. [30] showed that the hyperbolic model λ fits better than the exponential model and can be written as follows:(3)$$\lambda =-\frac{\partial d_{i}}{\partial r_{i}}=c_{i}\;k_{i}{\;r_{i}}^{-k-1}=\;\alpha_{i}\;{r_{i}}^{\beta_{i}}$$
where $\alpha_{i}$ and $\beta_{i}$ are determined by the video content. Because Equation (3) has two variables, $\alpha_{i}$ and $\beta_{i}$, they cannot be deduced directly. Li has developed an updating algorithm based on the collocated CTU values of α and β and $r_{i}=bpp$ (bits per pixel) using the following equations:(4)$$\lambda_{comp}=\alpha_{old}\;{bpp}_{real}^{\beta_{old}}$$

The encoded bits for an entire frame or a CTU are known after the encoding. $\alpha_{old}$ and $\beta_{old}$ are the previous frames or collocated CTU at the same hierarchical level.(5)$$lr=ln\;\frac{\lambda_{real}}{\lambda_{comp}}$$
where *lr* is the log ratio between the estimated λ from the allocated *bpp* and the actual λ after the encoding is performed and the actual *bpp* is known.

Then, from the log ratio, the α and β parameters that define the updated λ according to the video complexity are updated as follows:(6)$$\alpha_{new}=\;\alpha_{old}+\;\delta_{\alpha }\;lr\;\alpha_{old}$$(7)$$\beta_{new}=\beta_{old}+\delta_{\beta }\;lr\;\beta_{old}\;ln\;{bpp}_{real}$$
and $\delta_{\alpha }=0.1$, $\delta_{\beta }=0.05$ are constants that govern the updating rate. The *QP* is calculated from λ, where $c_{1}=4.2$ and $c_{2}=13.71$, by the following:(8)$$QP=c_{1\;}\ln\lambda +c_{2}$$

The authors of [33] have proposed an Optimal Bit Allocation (OBA) RC method. According to the OBA method, an RD estimation is proposed instead of an R–λ estimation for better *D* minimization. The authors argue that the original R–λ optimization method proposed for HEVC in [34] is not optimal, and the direct calculation of the Lagrange multiplier yields higher distortion. Using this RD method, an accurate estimate of QP achieves the RC target.

This method uses the Taylor expansion method to obtain a closed-form solution by iterating over the Taylor expansion. The iterative procedure converges quickly and requires no more than three iterations.

### 2.7. ROI Compression

ROI compression is a lossy video technique that favors certain image areas over others. The research described in [35] implemented skin tone filtering to distinguish between humans in a conversational video and the background. The video encoder adaptively adjusts the coding parameters, such as QP, search range, candidates for mode decision, the accuracy of motion vectors, block search at the Macro Block (MB) level, and relative importance of each MB. Finally, the encoder assigns more resources, bits, and computation power to the MB that resides inside an ROI and fewer resources to the MB that resides inside a non-ROI.

RD optimization for conversational video detects the key features of the faces and allows separation between the human faces and the background [36]. The ROI was located around the detected faces. Then, the Largest Coding Units (LCUs) containing parts of faces are searched more deeply during the LCU encoding process and assigned a lower QP. Non-ROI LCUs were searched for shallower parts and assigned higher QP values. The model uses a quadratic equation that determines the relationship between the desired QP, the bit budget, the Mean Average Difference (MAD) between the original LCU and the predicted LCU, model parameters, and the weighted ratio between the bits allocated for the ROI and those assigned for the non-ROI.

A novel ROI compression for the H.264 Scalable Video Coding (SVC) extension was proposed and studied [37]. The ROI is dynamically adjustable according to the location, size, resolution, and bitrate. The authors of [10] allocated the Lidar bitrate budget according to the ROI and non-ROI, significantly improving the quality of self-driving car depth maps. In [38], the encoder allocated the CTU bits according to their visual saliency. The QP for each CTU is then calculated using Lagrange multipliers.

An object-based ROI using an unsupervised saliency learning algorithm for saliency detection was proposed in [39]. This study detected salient areas in two steps: detection and validation.

The authors of [40] proposed ROI compression for static camera scenes by considering ROI blocks significantly different from the previous frame and applying the discrete Chebyshev transform (DTT) instead of the conventional discrete cosine transform (DCT).

The authors of [41] critique traditional ROI image compression, where separately predicting and coding the image is suboptimal. They propose a unified, deep learning framework that performs these tasks simultaneously in an end-to-end manner for direct rate-distortion optimization. The novel encoder network concurrently generates an implicit ROI mask and multi-scale image representations to guide the bit allocation process. This jointly optimized approach yields significantly better objective performance and visual quality compared to traditional cascaded methods. In [42], ROI rate control for H.265 video conferencing was solved by applying ideas from game theory. The encoder then allocates the ROI and non-ROI CTUs using Newton’s method.

### 2.8. HEVC Codec Applications

The most popular HEVC codec for research purposes is the HEVC Test Model (HM) reference software (latest release is HM 18.0) developed by the Joint Collaborative Team on Video Coding (JCT-VC). It is a reliable source for ensuring compatibility and adherence to the HEVC standards. Although its time performance improved from the HM-16.6 to the HM-16.12, the encoding must be faster and is still far from real-time requirements [43]. The x265 is among the most popular HEVC Codecs. It is continuously developed and improved by MulticoreWare and used or supported by many popular video applications such as FFmpeg, Handbrake, and Amazon Prime Video. It also has fast coding modes suitable for real-time applications. However, providing a δQP map for each image CTUs’ is not supported.

We selected the Kvazaar [32] HEVC Codec for this study. Kvazaar is an open-source academic video Codec specializing in high-performance, low-latency encoding using the HEVC (H.265) standard. It is suitable for real-time applications and offers a command-line interface and an API for flexibility. Kvazaar enables the definition of a δQP ROI map per image and CTU.

Teleoperation requires minimal G2G delay, necessitating low-delay video encoding. This can be achieved by limiting the encoded video GOP sequence to intra (I) and predictive (P) frames. Highly coding-efficient bi-directional (B) frames are not allowed here because their encoding process introduces an additional delay of several frames.

## 3. Closely Related Research Works

Several studies have explored region of interest (ROI)-based video compression and its applications, particularly in contexts relevant to autonomous vehicles and teleoperation. While these works provide a valuable foundation, they also reveal certain limitations that our current research aims to address.

The following thesis [44] explores the possibility of reducing data transmission by compressing a video stream based on the importance of specific features by using static and dynamic feature selection, with more critical components receiving less compression by segmenting the image into the ROI and region of non-interest (RONI). While this work mentions the concept of multi-category ROI compression by allowing traffic signs with lower *QP*, no practical demonstration is given for this use case, and it primarily focuses on a binary ROI/RONI distinction. Such a two-category approach, as our preliminary findings show, may not sufficiently enhance the quality of small, critical elements like traffic signs and lights when the general ROI is large.

Tong’s Ph.D. dissertation [45] proposed a ROI rate control for H.263 video conferencing, segmenting faces using skin color and motion vectors. First, the face in a video conversation is segmented from the image by skin color and the mosaic rule in the *I* frames, followed by motion vectors only for the *P* frames. Then, after the segmentation maps are created, ROI quadratic RC methods are applied to the H.263 video compression. This work, while pioneering for its time and application, is specific to face detection in video conferencing and utilizes older compression standards.

Bagwe’s master’s thesis [46] focuses on bandwidth saving for autonomous vehicles (AVs) by skipping redundant frames. While relevant to overall data reduction, this approach does not specifically address the perceptual quality of different regions within the transmitted frames, which is crucial for a remote operator’s ability to interpret the scene accurately.

The work by Hofbauer et al. [47] presents two techniques to improve the video quality for the remote driver. The first scheme focuses on adapting a traffic-aware multi-view video stream, beginning with traffic-aware view prioritization. This approach determines the importance of individual camera views based on the current driving situation. The second scheme reduces the bandwidth of a separate camera stream by applying dynamic elliptical masks over the ROI and blurring the area outside the ROI. This causes the video encoder to use fewer bits for the background image blocks. However, the elliptical masking and general blurring approach for a single stream, while reducing background bits, may not offer the fine-grained control needed to specifically enhance the legibility of diverse and irregularly shaped critical objects like textual traffic signs within the primary forward-facing view.

Skupin et al. [48] address rate assignment in a distributed tile encoding system in multi-resolution tiled streaming services based on the emerging VVC Standard. A model for rate assignment is derived based on random forest regression using spatio-temporal activity and encodings of a learning dataset.

In a preliminary study [49], it was demonstrated that a content-adaptive encoding scheme using semantic segmentation can improve video quality within a region of interest (ROI) for remote driving applications when compared to standard encoders. The authors noted, however, that the performance gains were limited due to their method’s simplistic partitioning of the video frame into only two regions: the ROI and the background.

Many existing ROI-based encoding methods, while improving general ROI quality, often fall short in scenarios requiring highly granular prioritization. For instance, a simple two-category (ROI/background) segmentation often fails to substantially improve the legibility of small but vital elements like traffic signs and lights when these are part of a larger, general ROI. This limitation underscores the need for a more nuanced approach. Our work seeks to address this gap by proposing a multi-category ROI compression scheme. This scheme specifically differentiates between ‘strong ROIs’ (e.g., traffic signs and lights, which are often difficult to discern under compression yet are paramount for safety) and ‘weak ROIs’ (other important driving elements like roads and pedestrians), alongside the background. By allocating bandwidth with greater granularity, particularly by giving strong ROIs a significantly higher share of bits relative to their size, our method aims to demonstrably improve the perceptual quality of these most critical elements, a challenge not fully resolved by the broader or less granular ROI techniques discussed in prior works.

## 4. Method: Three Categories of ROI Compression

In this work, we categorized the pixel classes into three separate categories. Category 2 contains the classes that are most important for the teleoperation of self-driving cars. These are traffic signs (standard and large) and traffic lights. During video compression, our goal is to compress CTUs containing pixels of such classes with pixel counts that exceed a certain threshold to the best video quality possible with our constraints of a specific low bandwidth and requirement for ultra-low delay. These classes are considered to have a strong ROI. Category 1 contains all the pixel classes that are also important for driving. Among them are the roads, lane marking of various types, cars, pedestrians, and bicycles. In our process of ROI compression, we do not want to degrade the quality of these objects compared to standard compression. The third, category 0, is the background classes. These include the sky, trees, vegetation, and buildings. These classes are least important for remote driving, and we can allow them to be transmitted with a lower quality.

The Cognata [9] driving simulator generates driving scenarios. Among the many sensors and maps that this simulator can simulate, we are interested in the semantic segmentation maps of each simulated video frame. Since the photo-realistic video image sequence and the related semantic segmentation images are generated by the same tool, the semantic segmentation maps are an almost perfect ground truth of their photo-realistic counterparts.

The division of the classes into three ROI categories that we used in this analysis is described in Table 1. It is only a specific proposal tailored to a particular driving environment. It may be varied in a different environment. For example, suppose there is no separation, such as a sidewalk, between the building and the road. In that case, it may be more practical to move the near-building pixel class from the background category to the weak ROI category, or if the acoustic wall is always well separated from the road, moving its class from the weak ROI to the background may be more effective in reducing background bandwidth.

Figure 2 explains the encoding process. Image (a) describes a non-compressed ‘png’ image as taken from a photo-realistic simulated driving scenario. The frame rate is 30 fps, and the resolution is HD of 1080 × 1920 pixels. Image (b) is the semantic segmentation map of (a). Each of the pixel classes has its unique color. Image (c) describes the grouping of the semantic map pixel classes into pixel categories according to Table 1. Background pixels that belong to the background classes are colored black, all pixels that belong to the weak ROI classes (classes that are important for teleoperation and road understanding) are colored gray, and the pixels that belong to the strong ROI containing the traffic signs and traffic lights are colored white. According to the HEVC standard [50], the image is divided into CTU rectangles of a constant size. In our case, the 1080 × 1920 HD image has 17 × 30 CTUs, each of 64 × 64 pixels. The last row of CTUs is 56 × 64 pixels in size. Image (d) describes the conversion from pixel categories into CTU assignments according to the following:(9)$$M_{CTU}^{i,j}=0\;If\;N_{CTU}^{i,j,2}>t_{2}\;M_{CTU}^{i,j}=2\;Elseif\;N_{CTU}^{i,j,1}>t_{1}\;M_{CTU}^{i,j}=1$$
where $M_{CTU}^{i,j}$ is the CTU category assignment for CTU at row *i* and column *j*. $N_{CTU}^{i,j,c}$ is the number of pixels of a CTU of category *c*. $t_{1}$ and $t_{2}$ are the threshold values for categories 1 and 2, respectively.

In this analysis, $t_{1}$ and $t_{2}$ were set to 512, which is one-eighth of the number of pixels in our CTU. We can observe in (c) that some of the small traffic signs that are colored white are converted into background CTUs because their pixel count of category two does not exceed the threshold value. This thresholding saves us bandwidth by not encoding with lower QP (higher bandwidth) CTUs that are far away and contain traffic signs, and traffic lights that are barely visible. Also, CTUs with only a few weak ROI pixels are assigned as background CTUs. Thresholding is also essential in real and not simulated video encoding, where a trained DNN produces the semantic segmentation maps, has segmentation errors, and can neglect a small number of pixels classified erroneously.

The next step is to take the CTU HD 17 × 30 category map of every image and convert it into δQP values. A category 2 CTU is assigned with δQP = −*q*, category 1 with δQP = 0 (no change from the base QP), and category 0 with δQP = *q*. The value *q* is an integer value between one and ten. The video encoder does not allow for higher values due to stability issues and the over consumption of bandwidth. The higher the δQP, the higher the DCT/DST quantization, resulting in a coarser image and reduced bandwidth. The non-compressed movie sequence of ‘png’ images was converted into a YUV420p raw video by FFmpeg [51], the δQP maps of 17 × 30 values, one map for every image was represented in binary, and both raw video and ROI description were fed into the Kvazaar HEVC Codec. The bitrate was set to 1 Mbps, and low-delay operation mode was selected. The RC that was used is a frame-level λ domain rate control that adjusts the frame λ when advancing along the video frames according to frame bitrate undershoot or overshoot. Then, the base frame ${QP}_{b}$ is calculated as a function of λ. The actual QP of each CTU is then calculated by the following:(10)$$QP=QP_{b}+\delta QP$$

To avoid out-of-range values, clipping should be performed to the HEVC *QP* range:(11)QP=CLIP(0,51,QP)

The video sequence was also compressed by Kvazaar without an ROI map. In the former case, the rate-control method that is applied is an implementation of Li’s [30] λ domain rate control, where the rate control is also optimized over the individual CTUs.

The metrics we use to evaluate the performance of the ROI compression over the three ROI categories are based on the PSNR and SSIM. Since we are interested in the MPSNR (mean PSNR) concerning the three different ROI categories, we define the MPSNR by the following:(12)$${MPSNR}^{c}=\frac{1}{N^{c}}\sum_{\begin{array}{c} Over\;all\;CTUs\prime \\ PSNR\;of\\ category\;c \end{array}}{PSNR}_{CTU}^{i}$$
where $N^{c}$ is the number of CTUs of category *c* and ${PSNR}_{CTU}^{i}$ is the PSNR of the *i*’th CTU. Similarly, we define the categorical MSSIM (Mean Structural Similarity Index) by the following:(13)$${MSSIM}^{c}=\frac{1}{N^{c}}\sum_{\begin{array}{c} Over\;all\;CTUs\\ MSSIM\;of\\ category\;c \end{array}}{MSSIM}_{CTU}^{i}$$

## 5. Results: Simulation of Driving Case Studies with the Multi-Category ROI Method

### 5.1. Analysis of a Simulated Driving Movie Clip Containing Traffic Signs as Well as Textual Signs

The following comparison between the standard Kvazaar and ROI compression is from a Cognata-simulated driving movie from Lombard Street in San Francisco. Figure 2a is an image taken from the simulated movie clip. (b) describes its semantic segmentation result, where each class is uniquely colored. (c) describes the categorization of the pixel classes into three categories as described in Table 1. The white is for category 2, the gray is for category 1, and the black is for category 0. (d) describes CTU thresholding according to Equation (9). Both compression results are shown in (e) and (f). (e) is an image taken from a compressed video sequence by standard compression, and (f) is from the ROI compression. Our observations reveal a notable enhancement in the visual quality of the textual sign displaying the message “TRAFFIC FROM LEFT DOES NOT STOP” with ROI compression. Additionally, there is a discernible improvement in the visual clarity of the stop sign. It is worth noting that common traffic signs are engineered to maintain visibility even under adverse conditions such as inclement weather, wear and tear, and deliberate damage, characteristics that also extend to poor video quality. We can observe that the quality of the weak ROI objects, such as roads, road marks, and sidewalks, stays roughly the same and even improves slightly. At the same time, there is a significant degradation in the quality of the background objects. Meanwhile, the sky, buildings, and trees appear blockier than in the standard video. (g), (h), and (i) show the textual and stop signs for the original non-compressed, standard, and ROI-compressed images. Images (j), (k), and (l) are the enlargements of the textual sign. A remote driver that observes this textual sign by standard video can hardly read the sign, while by applying ROI compression, the sign is well readable. 

Our analysis is further supported by Appendix A, which provides a dynamic visual comparison of video quality between standard HEVC encoding and our proposed three-category Region-of-Interest (ROI) HEVC encoding. Both encoded videos in this Appendix A are presented at a consistent target bitrate of 1 Mbps. The video display is structured into four distinct quadrants: the upper left showcases the original semantic map video, while the upper right displays the ROI categorization map before performing CTU thresholding as described in Equation (2), where regions are clearly classified as strong ROI (white), weak ROI (grey), and background (black). The lower left quadrant presents the video encoded using standard HEVC, and the lower right reveals the video processed with our three-category ROI HEVC method. This Appendix A vividly demonstrates the enhanced perceptual quality achieved within strong ROI areas by our proposed approach when directly compared to standard encoding at equivalent bitrates. It is important to note that brief pauses have been intentionally introduced after the appearance of each traffic sign within the video; these pauses are solely to allow viewers ample time for visual comparison between the two encoding methods and are not part of the original video sequence.

Figure 3 describes traffic signs taken from different frames of the same driving video simulation. Column (a) shows the non-compressed traffic signs taken from the PNG images. Column (b) describes the compression results by the standard method, while column (c) describes the traffic signs that appear after the ROI compression. The traffic signs in column (c) seem much closer to their original images than their standard compression images.

Figure 4 compares image quality metrics between the three ROI categories of the standard and ROI compression. Figure 4a compares the ROI categorical PSNR of both video compression methods. The PSNR of every category is calculated by averaging over all its individual CTU PSNR of the same ROI category. The length of the movie sequence was 600 images, but for clarity, only part of the sequence is shown here. The means of the metric are calculated over the whole image sequence. The dashed lines describe the PSNRs for the three categories of background, weak ROI, and strong ROI, and the contiguous lines represent the three categories of PSNRs for the ROI compression. We can observe that the blue lines that describe the background PSNR move on average 3 dB downward when switching from the standard to ROI compression, leaving more bandwidth in the strong ROI category. This change can also be reflected when comparing the sky, trees, and building background categories, as shown in Figure 2e,f, where there is a significant degradation in background quality. Comparing the yellow lines shows a slight improvement in the weak ROI by applying ROI compression. In this case, it is 0.7 dB, but in general, we do not want to degrade the quality of the weak ROI that contains classes that are important for remote driving, but any figure that is not negative can be acceptable. Comparing the two red lines shows the improvement in the strong ROI CTUs. In this example, the gain is more than 5 dB and is very noticeable when looking again at Figure 2e,f.

Figure 4b compares both compression methods’ categorical Mean Structural Similarity Index (MSSIM). The MSSIM is a variation of the SSIM that calculates the mean value of the SSIM across local image regions, providing a more global measure of image similarity [52]. Here, we average the SSIM of all CTUs of a specific category, similar to the categorical PSNR calculation. The MSSIM metric for the background (blue dashed and contiguous lines) degrades from the mean overall images of 0.85 to 0.79, the weak ROI improves slightly from 0.75 to 0.77, and the strong ROI that contains the traffic signs improves significantly from 0.75 to 0.9, enabling an easy reading of the textual traffic sign during driving.

As expected, the overall quality metrics for the video sequence that is ROI-compressed should be lower than those of standard compression, which does not prioritize any region in the image. Figure 5a shows the movie clip image PSNR in blue for standard compression and yellow for ROI compression. The standard compression PSNR is almost always higher than the ROI compression, since the overall PSNR can be optimized by minimizing the error variance among the CTUs. When comparing the average PSNR of the standard compression to that of the ROI compression, we lose 1.1 dB. Figure 5b shows the video SSIM for both standard and ROI compression. Again, the image metric is degraded. The SSIM is reduced from 0.8 for the standard compression to 0.78 for the ROI compression. This comparison is essential as a preliminary logical check that the ROI compression method should degrade the overall metrics of the video motion sequence. However, there is a considerable benefit in increasing the quality of the ROI, particularly the strong ROI.

### 5.2. Analysis of a Simulated Driving Movie Clip Containing Traffic Lights

The second example is from a simulated driving clip in Munich, Germany, containing traffic lights. It is described in Figure 6. Image (a) illustrates the non-compressed image with traffic lights. Image (b) is the semantic segmentation of the original image. The traffic signs are pinkish-orange, and the lights are pale purple or lavender. (c) describes the categorization of every pixel to the background (black), weak ROI (gray), and strong ROI (black) according to Table 1. (d) shows the result after applying the thresholding as described in Equation (9). (e) and (f) are the standard and ROI compression results. Again, the sky, buildings, and trees degrade in quality, making their images blurry and bulky. The weak ROI of roads, cars, and sidewalks remains at least with their standard compression quality. The traffic lights, as well as traffic signs, significantly improve their visible quality. (g) to (i) describe the enlargement of the traffic light with a ‘Front and right’ traffic sign above and ‘Main Road’ traffic sign below. Again, the traffic lights and signs are transmitted with much better quality on account of the background’s bandwidth, which contains the sky, buildings, and low vegetation, which are less critical for the remote driver. Figure 7a describes the categorical PSNR comparison between the standard and ROI compression. The red lines represent an improvement of 6.5 dB on average in the CTUs containing traffic lights or traffic signs. There are missing values in the red curves because not all images have traffic signs or traffic lights that passed the required pixel count threshold. The yellow lines, describing the weak ROI, show almost no change, and the blue lines indicate a 3 dB reduction.

Figure 7b shows a similar comparison but for the categorical MSSIM metric. The strong ROI improves on average from 0.75 to 0.9, the weak ROI improves slightly from 0.75 to 0.77, and the background reduces from 0.85 to 0.79.

## 6. Discussion

### 6.1. Compare the Three ROI Categories with the Two-Category Method

The clear advantage of a three-category ROI compression over only a two-category ROI is shown in this section. The visibility of a textual traffic sign is compared among the compression methods that are described in this work. The movie clip we analyze is the same as defined in Section 4. All compression methods were compared at a 1 Mbps bandwidth. Figure 8a,b show the same movie frame. (a) was taken from the two-category ROI compressed movie clip, while (b) was taken from the three-category ROI. It can be noticed that both are ROI-compressed by a visual comparison of the sky blockiness to that of the road blockiness for both frames. The general quality of both images is similar, except for the quality of the traffic sign area, which significantly improves. Rows (c), (d), and (e) describe the traffic sign along the driving path at three different locations.

The four columns of rows (c) to (e) describe (from left to right) the original non-compressed ‘PNG’ images, standard compression, two-ROI, and three-ROI compression. Results show that categorizing the image classes into only the background and ROI is far from enough to make the textual traffic sign readable. Separating the original ROI into at least a weak and strong ROI is necessary.

Table 2 shows the PSNR/MPSNR and MSSIM metrics for comparing the three compression methods. Metrics were calculated by averaging all images in the movie clips. As expected, the overall metrics degrade by more than 1 dB for the PSNR and 0.02–0.03 for the MSSIM, the background quality degrades significantly, while the weak ROI improves slightly for both methods. In contrast, in the strong ROI region, there is a slight improvement for the two categories of ROI compression and a vast improvement for the three-category method.

### 6.2. Analysis Across Various Bandwidths and q Values

We analyzed three bandwidths (1, 2, and 4 Mbps) and three q values (3, 5, and 10) to assess the visibility enhancement for traffic signs and lights. Actual bandwidths are also provided alongside the target values to ensure the bandwidth is consistently maintained, even for ROI compression. The results of the PSNR/MPSNR and SSIM/MSSIM are described in Table 3. The rows representing the visibility of traffic signs and lights designated without applying *q* (NA) describe the default Kvazaar RC results without using ROI techniques. Comparing the MPSNR for the strong ROI shows a significant improvement for all bandwidths, up to 6.5 dB for the 1 Mbps, 7.6 dB for the 2 Mbps, and 8.4 dB for the 4 Mbps. The quality of the weak ROI never degrades and even improves up to 1 dB depending on the bandwidth and *q*. Improvement in ROI quality is due to the migration of the bandwidth budget from the background, which loses up to 1.3 dB, depending on the bandwidth and *q*.

### 6.3. Limitations

ROI compression is effective only if there is enough bandwidth to be transferred from the background area to the ROI. Therefore, the background should be a substantial portion of the image and of considerable complexity. This is not always true, but in many cases, the vehicle’s camera view contains significant background areas, such as the sky and vegetation in rural areas or buildings in urban areas.

Separating the ROI into multiple categories of importance, or at least into weak and strong ROI, relies on the condition that the strong ROI area is small. This may not be true in crowded urban areas with many traffic lights and signs. This method is effective when the bandwidth allocated for the CTUs that contain the traffic signs and traffic lights can be increased by an order of magnitude, and this is possible in many cases where the image includes a few close traffic signs and lights.

This research relied on artificial movie clips generated from a photo-realistic driving simulator. The semantic segmentation ground truth was also available. We must use non-compressed video sources and real-time DL semantic segmentation methods in a realistic environment, not a simulated one. Such systems exist today. One example is the SOTA real-time image segmentation technique described in [53]. Its small PIDNetS model can reach 90 FPS with the Nvidia 3090 GPU at an mIOU (mean intersection over union) accuracy of around 79%. The deficiencies of the practical semantic segmentation methods from the ground truth can be mitigated by applying our CTU ROI assignment thresholding method.

### 6.4. Additional Computer Vision Disciplines That May Benefit from Multi-Category ROI

The multi-category region of interest (ROI) methodology, which enhances the perceptual quality of critical objects by reallocating bandwidth from less important background elements, has significant potential beyond autonomous driving applications. The core principle of segmenting a scene into tiered categories of at least a strong ROI, weak ROI, and background can be adapted for fields such as medical imaging, remote robotics, security surveillance, and precision agriculture. A particularly relevant application is video conferencing. For video conferencing, the goal shifts from identifying traffic signs to preserving the fidelity of human connection, especially over networks with limited or variable bandwidth. This aligns with early ROI research that focused on conversational video. By segmenting the video feed based on communicative importance, the user experience can be dramatically improved without increasing bandwidth requirements. The video conferencing categories for its pixel classes may be the following:Strong ROI: This category would consist exclusively of the regions containing the speaker’s eyes.

Rationale: The rationale for categorizing the eyes for the highest importance is that they are critical for conveying emotion, attention, and sincerity. Clear, crisp eyes create a stronger sense of presence and eye contact, which is vital for building trust and engagement in a virtual setting. By dedicating the most bandwidth to this small but crucial area, the system ensures these non-verbal cues are never lost to compression artifacts.Weak ROI: (Medium Priority): The face and upper body. This category would include the rest of the face (mouth, nose, facial structure), as well as the speaker’s upper torso and hands if they are in the frame.

Rationale: The mouth is essential for lip-reading and complements the audio track. Broader facial expressions and hand gestures are also key components of non-verbal communication. This region provides the necessary context for the “Strong ROI.” The goal is to preserve the quality of these elements, ensuring they are not degraded compared to standard compression.Background: This category includes everything else: the user’s room, virtual backgrounds, office furniture, etc.


## 7. Conclusions

In summary, this research has focused on refining and evaluating a region of interest (ROI) compression method within the context of autonomous driving scenarios. We utilized the Cognata proprietary driving simulator to generate realistic driving scenarios, with access to ground truth semantic segmentation maps. The segmentation maps contain approximately 50-pixel classes, each uniquely colored, representing various elements in the simulated environment. We categorized these pixel classes into three distinct ROI categories: strong ROI, weak ROI, and background. The strong ROI included crucial elements like traffic signs and lights, while the weak ROI encompassed important driving features such as roads, lane markings, vehicles, and pedestrians. Background classes represented less critical elements like the sky, trees, vegetation, and terrain. Our approach involved applying ROI-based compression, with different quantization levels for each category, to prioritize the quality of critical driving elements while conserving bandwidth. The process involved converting pixel categories into CTU assignments based on pixel count thresholds, ensuring that strong ROI elements received the highest compression quality. We conducted an extensive analysis using simulated driving scenarios. We evaluated the image quality metrics of PSNR and MSSIM for the different ROI categories. The results revealed significant improvements in the quality of the strong ROI elements, particularly traffic signs and traffic lights, while at least maintaining the quality of the weak ROI elements. The new multi-category ROI compression method shows a clear advantage over a ROI compression that partitions the video into only the ROI and background.

## 8. Future Work

While this study successfully demonstrates the objective benefits of a multi-category ROI compression framework, several avenues for future research are apparent. First, a crucial next step is to conduct a task-based subjective evaluation with human teleoperators. Such a study would measure the real-world impact of our method on driver performance, reaction time, and cognitive load, providing a more definitive assessment of its effectiveness than objective metrics like PSNR and SSIM alone.

Second, a detailed analysis of the temporal dynamics of quality variation is warranted. Future work should investigate methods to ensure temporal stability, minimizing perceptible quality fluctuations or flickering between the ROI categories as they change size and position, which can be detrimental to the operator’s visual experience and development of accurate, practical methods for varying the baseline *QP*.

Finally, while this work utilized a simulator with perfect ground-truth segmentation, future research will focus on integrating our compression framework with a state-of-the-art, real-time semantic segmentation model operating on real-world video feeds. This will involve addressing the challenges of segmentation errors and ensuring the end-to-end system remains robust and efficient for deployment in live teleoperation scenarios.

## Figures and Tables

**Figure 1 entropy-27-00674-f001:**
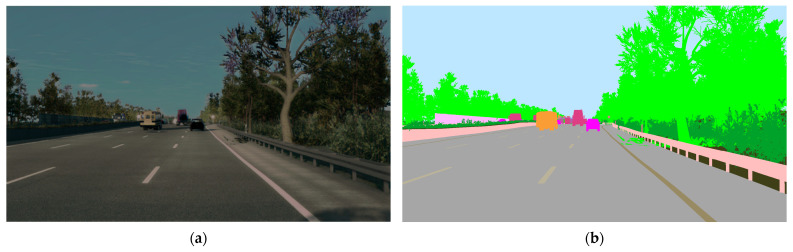
(**a**) Snapshot of a driving scenario generated by Cognata in PNG (lossless) format; (**b**) semantic segmentation of the original image according to the classes described in Table 1. Every color represents a different class.

**Figure 2 entropy-27-00674-f002:**
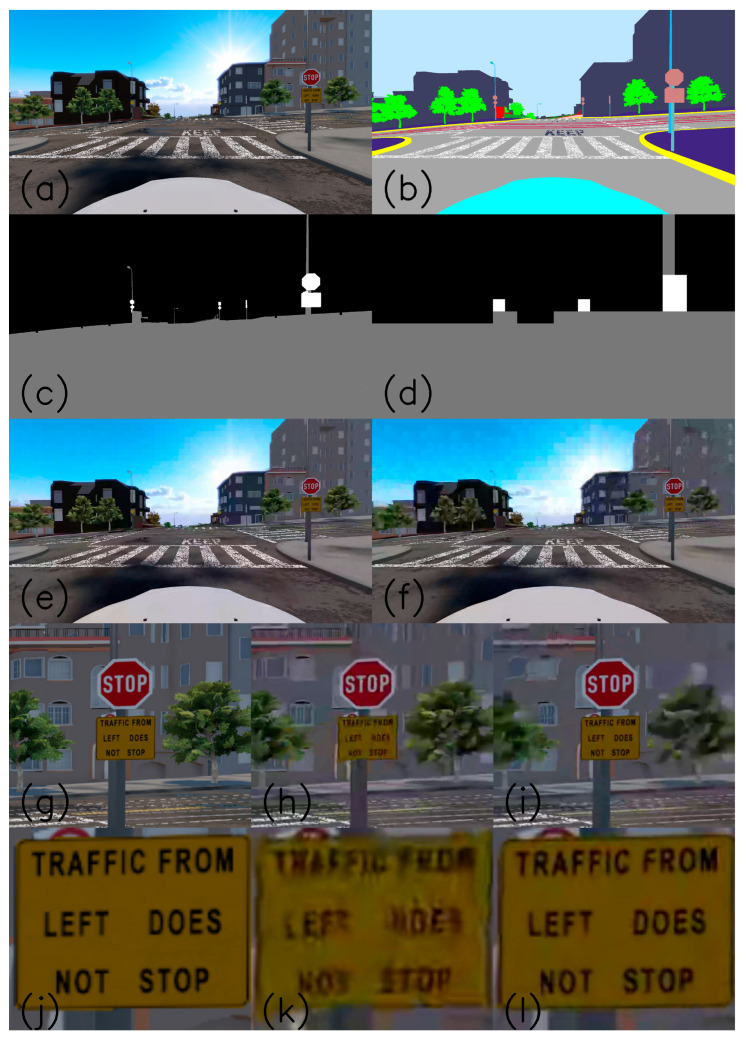
Description of the three categories of the ROI video compression method. The video bandwidth is 1 Mbps. (**a**) is the original non-compressed image. (**b**) is the semantic segmentation of the original image. (**c**) describes the grouping of the semantic classes that are visualized in (**b**) into background (black), ROI (gray), and strong ROI (white). (**d**) is the quantization into CTU boundaries and application of the pixel count threshold. (**e**) is the standard compressed image. (**f**) is the compressed ROI. (**g**–**i**) are the enlarged traffic sign areas as they appear in the original, standard, and ROI-compressed images. (**j**–**l**) are enlargements of the textual traffic sign.

**Figure 3 entropy-27-00674-f003:**
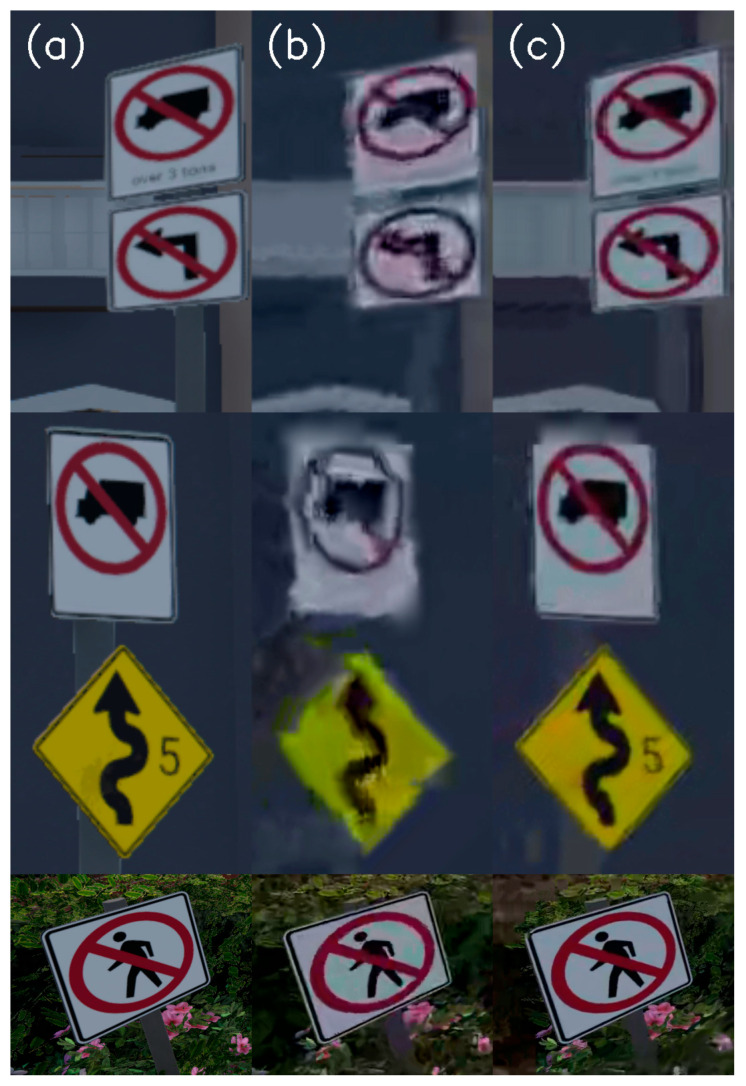
Traffic signs were taken from the same simulated driving video, but from other frames. (**a**) describes traffic signs taken from the non-compressed images. (**b**) describes traffic signs as compressed by the standard method. (**c**) describes the traffic signs that are compressed by the ROI method.

**Figure 4 entropy-27-00674-f004:**
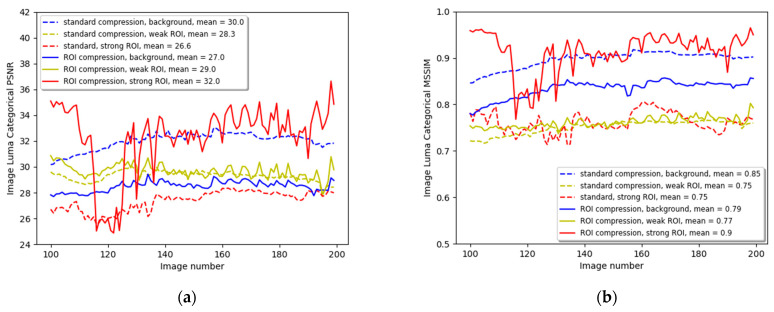
(**a**) The categorical image sequence luminance PSNR. The dashed lines describe the standard compression background, weak ROI, and strong ROI, while the contiguous lines describe the ROI compression PSNRs; (**b**) the categorical image sequence luminance MSSIM.

**Figure 5 entropy-27-00674-f005:**
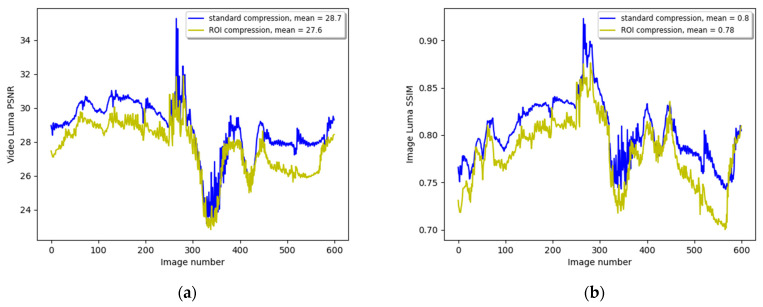
(**a**,**b**) The overall video luminance PSNR and SSIM for standard and ROI compression, respectively.

**Figure 6 entropy-27-00674-f006:**
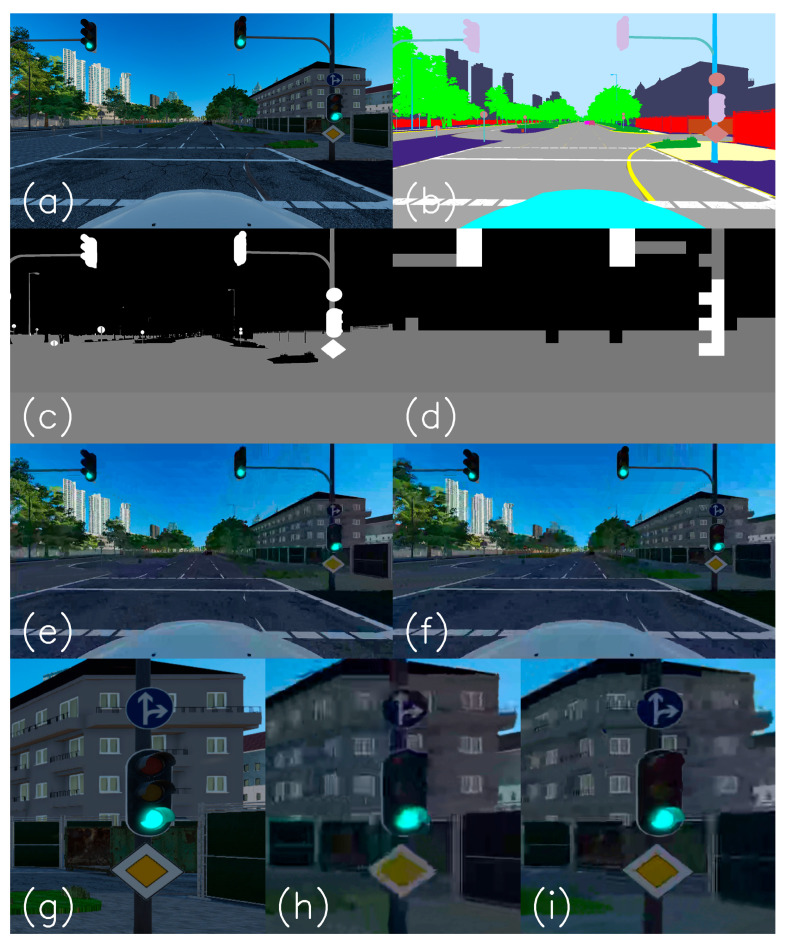
Description of the three categories of ROI video compression methods at 1 Mbps. (**a**) is the original non-compressed image. (**b**) is the semantic segmentation of the original image. (**c**) is the categorization of the semantic classes. (**d**) is the CTUs assignment. (**e**) is the standard compressed image. (**f**) is the compressed ROI. (**g**–**i**) are the enlarged traffic lights and traffic signs surroundings as they appear in the original, standard, and ROI-compressed images.

**Figure 7 entropy-27-00674-f007:**
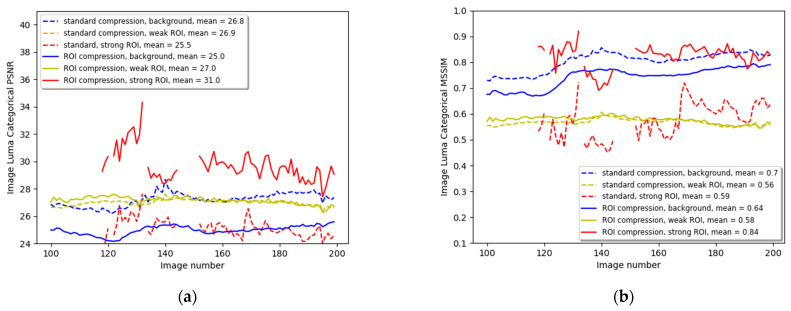
The categorical image sequence luminance PSNR (**a**) and MSSIM (**b**), respectively, for the Munich driving video clip. The dashed lines describe the standard compression background, weak ROI, and strong ROI, while the contiguous lines describe the ROI compression PSNRs and MSSIMs. There are missing values in the red curves because not all images have traffic lights or signs of significant size.

**Figure 8 entropy-27-00674-f008:**
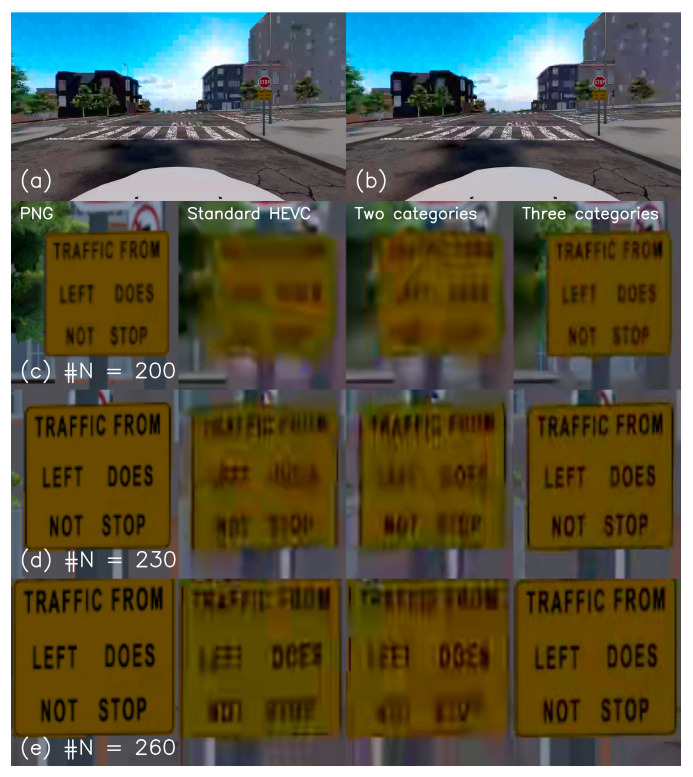
Comparison of the video encoding methods. (**a**,**b**) describe the three and two categories of ROI compression results. (**c**–**e**) The four columns from left to right are the original PNG non-compressed images, the standard HEVC compression results, the two categories of ROI, and the three categories of ROI.

**Table 1 entropy-27-00674-t001:** Categorization of the image segmentation classes into 0 (background), 1 (weak ROI), and 2 (strong ROI) pixel categories.

Category of Pixel Classes	Pixel Classes
2 (most relevant, strong ROI)	Traffic Sign, Large Sign, Traffic Light
1 (relevant, weak ROI)	Road, Parking, Railway, Lane marking, Road marking shape, Sidewalk, Biking Lane, Car, Truck, Bus, Motorcycle, Drone, Rider, Pedestrians, Bird, Curb, Tunnel, Bridge, Fence, Guardrail, Acoustic wall, Props, Electricity cable, Pole, Electric light pole, Ego Car, Bicycle, Van, Lane Line Types: Dashed, Solid, Double Solid, Double Dashed, Double Solid and Dashed, None, Animal, Gantry, Trailer, Personal Mobility, Construction Vehicle, Rock
0 (background)	Unlabeled, Tree, Low vegetation, Sky, Building, Far Building

**Table 2 entropy-27-00674-t002:** PSNR/MPSNR and MSSIM comparison between standard, three, and two ROI video encoding.

Compression Method	Overall PSNR/MSSIM	MPSNR/MSSIM per Category		
		Background	Weak ROI	Strong ROI
Standard	28.7/0.8	30.0/0.85	28.3/0.75	26.6/0.75
Two categories	27.4/0.77	26.6/0.77	29.3/0.78	27.3/0.77
Tree categories	27.6/0.78	27.3/0.79	28.9/0.77	33.1/0.9

**Table 3 entropy-27-00674-t003:** PSNR and MPSNR analysis for the Lombard driving clip at various bandwidths and *q* values. NA is not applicable.

Band Width (Mbps)		*q*	Overall PSNR [db]/MSSIM	MPSNR per Category [db]/MSSIM		
Target	Actual			Background	Weak ROI	Strong ROI
1	1.008	NA	28.7/0.8	30/0.85	28.3/0.75	26.6/0.75
1	1.016	3	28.7/0.81	29/0.84	29/0.77	28/0.82
1	1.01	5	28.5/0.8	28.9/0.82	28.9/0.77	29.8/0.86
1	1.006	10	27.6/0.78	27.3/0.79	28.9/0.77	33.1/0.93
2	2.008	NA	30.3/0.84	31.5/0.85	29.8/0.75	28.4/0.75
2	2.012	3	30.3/0.84	30.9/0.83	30.4/0.77	30.7/0.87
2	2.012	5	30.1/0.84	30.2/0.86	30.6/0.82	32.3/0.92
2	2.006	10	29.1/0.82	28.5/0.82	30.7/0.82	36/0.96
4	4.003	NA	32.1/0.88	33.1/0.92	31.5/0.84	30.3/0.88
4	4.008	3	32/0.88	32.4/0.91	32.2/0.85	33.1/0.93
4	4.004	5	31.8/0.88	31.8/0.89	32.4/0.86	34.8/0.95
4	3.995	10	30.8/0.86	29.9/0.85	32.6/0.86	38.7/0.98

## Data Availability

The data presented in this study are available on request from the corresponding author.

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
