# Peer review of "Improved Perceptual Quality of Traffic Signs and Lights for the Teleoperation of Autonomous Vehicle Remote Driving via Multi-Category Region of Interest Video Compression"

_entropy, 2025, doi:10.3390/e27070674_

Round 1
Reviewer 1 Report
Comments and Suggestions for Authors
- The paper currently assumes that recognising traffic signs and lights are the most important perceptual tasks facing the remote operator of an autonomous vehicle. This may well be true for the purposes of this paper, which is focused on providing the clearest information to a remote driver as possible in a severely time-constrained setting. However, the current text implies, for example, that recognising the text on a traffic sign is more important than spotting a pedestrian about to cross the vehicle’s path. Without providing a convincing argument, this comes across as a somewhat shaky assumption. A clear and concise argument is required, which is set in the context of teleoperation. This does not necessarily need to be highly complex, for example, it may have been shown that remote operator recognition of traffic light states and text on signs is the most difficult information to be provided in a timely and accurate enough manner to a remote vehicle operator and is therefore, in the context of this paper, the most important ROI category.
- Line 19: Be careful that you are not giving the message that your method is constrained to simulation. It is only the current testing reported in this paper that is conducted in simulation isn’t it?
- Line 24: “and most importantly”. This is currently an insufficiently justified statement, see first comment above.
- Line 27-30: The current text on these lines does not get the message across clearly enough. You have developed a method that swaps lower fidelity in compression of the relatively less critical background scene for increased fidelity in critical ROIs such as traffic lights and so on, whilst retaining the same fidelity in weaker ROIs, that are still important areas of the scene. This overall message currently only really comes out gradually as the reader progresses through the paper; therefore, it needs to be said explicitly and simply up front.
- Line 39: “This problem is much more complicated than initially thought by the designers.” should be “This problem is much more complicated than initially thought by many designers.” or similar. Although many may not have, some designers did foresee how hard this might be.
- Line 54-55: This claim is made in [3]-[5]? If so state this, otherwise it looks like an unsupported number plucked from the air.
- Line 55-56: The claim made in line 54-55 does not seem to lead logically to that made here without some further explanation. If the study mentioned thereafter is, somehow, that justification, then the sentences of this paragraph need reordering and sorting out, because the argument does not flow logically as it is presented currently.
- Line 93: “The most critical objects for driving, which are the traffic signs and lights”. See the first comment above. Currently, this statement lacks sufficient justification.
- Line 182-183: “The Cognata pixel classes are based on earlier works.” … whose earlier work?
- Lines 330-353: you describe some work which is closely related to yours which is fine and necessary, but it is much more useful to use a section such as this to also inform the reader about the weaknesses in the prior work of others that you will be addressing in the work following in this paper. This also serves as a means by which you can give a high-level description of your paper’s novel contribution.
- 2: The meaning conveyed by (c) and (d) is very unclear. You need to explain what is being illustrated more clearly.
- Line 408-409: This needs more explanation because, as mentioned, the meaning of (c) is unclear.
- Section 5 (and possible elsewhere): You need to be very careful in section 5 (and please check elsewhere) to make sure that when you are referring to a single compression method with multiple categories, that you decide on a term and stick with it, e.g. you could hyphenate, as you have done on line 570 (“a three-category ROI compression”) and always use the same term (e.g. line 579 uses “two ROI” and “three ROI”), so as to carefully distinguish between text where you are comparing between multiple methods of compression, as in line 584-585 (“comparing the three compression methods”). Otherwise, there is room for reader confusion, e.g., where on line 588-589 you say “there is a slight improvement for the two categories of ROI compression” I think you are referring to the two-category algorithm rather than the two methods themselves.
- Line 599-560: Table 2 seems to be strangely placed on the page for the version I have. Looking back, I notice that this is the case also in some earlier cases, (e.g. Fig. 1, Table 1 and others).
- Line 658-661: You should note that this is achieved at a cost to quality in the background category, which you believe represents a worthwhile improvement overall … or similar such text.
- Line 13: abbreviation “HEVC”?
- Line 13: Is this preliminary study cited elsewhere?
- Line 38: “unpolite” should be “impolite”.
- Line 54: “his” should be “his/her” or similar gender-neutral term.
- Line 119: “his” should be “his/her” or similar, again you must respect gender-neutrality.
- Line 139-141: “The goal for attaining the cost-effectiveness of such a hybrid system is to reach the ratio between almost fully automated cars on the road and human teleoperators in a single moment, which is around 1000 [13].” This sentence does not currently make grammatical sense, e.g., what is the meaning of the number 1000? I think, from reading the following sentence, that you mean a ratio of 1000:1 perhaps?
- Section 1.2, i.e., line 132-151: The paper needs reorganising so that application-oriented text appears much earlier so as to help achieve making the overall focus of this paper much clearer from the outset.
- Line 213: “CTU” is only written in full in the Abstract. Please state it in full the first time of use in the main text.
- Line 241: “The granularity of Li's algorithm” who is Li? It does not relate to the closest citation.
- Line 244: “The R-Lamba method is implemented in Kvazaar” similarly, who is Kvazaar? Citation not provided until line 321.
- Line 251: “compared” should be “compare”.
- Line 251: “Li et al.”, no citation provided, and there seem to be multiple authors in the references sections with this surname.
- Line 254-255: “Error! Reference source not found.)”? Please correct.
- 2: The subfield lettering is not clear, although it is much better in all other later figures.
- Line 450: “describe” should be “describes”. Same for lines 451, 453 and 533.
- Line 584: the table number is missing (Table 2 perhaps?).
- Line 592: “tree”?
Author Response
We thank the reviewer for the thorough and constructive feedback. The comments have been invaluable in improving the clarity, accuracy, and overall quality of our manuscript. We have carefully considered each point and have revised the paper accordingly.
Below, we provide a point-by-point response to the comments and suggestions.
Response to Comments and Suggestions for Authors
Comment 1: The paper currently assumes that recognising traffic signs and lights are the most important perceptual tasks facing the remote operator of an autonomous vehicle. This may well be true for the purposes of this paper. However, the current text implies, for example, that recognising the text on a traffic sign is more important than spotting a pedestrian about to cross the vehicle’s path. Without providing a convincing argument, this comes across as a somewhat shaky assumption.
Response 1: Thank you for this crucial observation. We agree that our original wording was misleading. The safety of pedestrians, cyclists, and other vulnerable road users is of paramount importance and is in no way secondary to recognizing traffic signs.
Our rationale for focusing on traffic signs and lights as the "strong ROI" is that they are often small, information-dense objects that are particularly susceptible to compression artifacts, making them one of the most difficult classes of objects for a remote operator to interpret accurately under the strict bandwidth and latency constraints of teleoperation. The proposed method provides a clear benefit by allocating a significantly larger bit budget to these challenging objects, which is effective due to the logarithmic relationship between bandwidth and video quality.
We have revised the manuscript throughout to remove any phrasing that implies a hierarchy of importance for safety. Instead, we now frame the problem in terms of recognition difficulty under compression.
Comment 2: Line 19: Be careful that you are not giving the message that your method is constrained to simulation. It is only the current testing reported in this paper that is conducted in simulation isn’t it?
Response 2: Thank you for this important point. We agree that the research is not constrained to simulation. The use of the Cognata photo-realistic simulator was a deliberate choice for this stage of research, as it provides a controlled environment with perfect ground-truth semantic segmentation data. This is ideal for validating the core compression methodology without the confounding variable of segmentation errors from a real-time deep-learning model. We even tried to apply our method with real non compressed driving video clips of ~3 GBytes/ 30 sec clip which we acquired in many driving situations. Unfortunately, the available CityScape models that work very well on their test set did’nt perform well on our original data. Due to different urban and lightining conditions. Offcouse this is solvable but at a cost of human tagging of thousands of carefully selected images out of our many hours of recorded non compressed YUV420p driving videos. This is beyond our budget but is possible for vehicle manufacures.
We have revised the text in the abstract and the "Limitations" section to emphasize that the simulation is a platform for validation and to discuss the clear path toward real-world implementation using state-of-the-art, real-time semantic segmentation systems.
Comment 3: Line 24: “and most importantly”. This is currently an insufficiently justified statement, see first comment above.
Response 3: We agree. Consistent with our response to Comment 1, this phrase and similar statements suggesting a hierarchy of importance have been removed or rephrased throughout the manuscript.
Comment 4: Line 27-30: The current text on these lines does not get the message across clearly enough. You have developed a method that swaps lower fidelity in compression of the relatively less critical background scene for increased fidelity in critical ROIs such as traffic lights and so on, whilst retaining the same fidelity in weaker ROIs... this needs to be said explicitly and simply up front.
Response 4: Thank you for this valuable advice on improving clarity. We have revised the abstract and introduction to state our contribution more explicitly. The text now clearly explains that our multi-category ROI framework significantly enhances the perceptual quality of the most critical objects (strong ROI) by reducing the fidelity of non-essential background elements, while ensuring the visual quality of other essential driving-related classes (weak ROI) is at least maintained.
Comments 5 & 6: Line 39: “This problem is much more complicated than initially thought by the designers.” should be “This problem is much more complicated than initially thought by many designers.” or similar. Although many may not have, some designers did foresee how hard this might be.
Response 5 & 6: We agree with this correction. It is more accurate to acknowledge that not all designers were overly optimistic. The text has been changed from "the designers" to "many of the designers" to reflect this.
Comments 7 & 8: Line 54-55: This claim is made in [3]-[5]? If so state this, otherwise it looks like an unsupported number plucked from the air. Line 55-56: The claim made in line 54-55 does not seem to lead logically to that made here without some further explanation.
Response 7 & 8: Thank you for identifying the need for a stronger and more current reference. The paragraph has been substantially revised. We have replaced the previous text with recent experimental results from a study on a 5G-enabled teleoperated vehicle, which reported an average Glass-to-Glass (G2G) delay of 202 ms and concluded that this was safe only for low-speed driving. This provides a clear, empirically supported context for the low-latency requirements of teleoperation.
Comment 9: Line 93: “The most critical objects for driving, which are the traffic signs and lights”. See the first comment above. Currently, this statement lacks sufficient justification.
Response 9: We agree. As per our response to Comment 1, the phrase "the most" has been removed. The text has been modified to refer to traffic signs and lights as "Critical objects for driving" or "Difficult-to-recognize classes".
Comment 10: Line 182-183: “The Cognata pixel classes are based on earlier works.” … whose earlier work?
Response 10: Thank you for this question. We have clarified this by explicitly mentioning the Cityscapes dataset as a key example of influential earlier work that is notable for urban scene understanding and upon which simulators like Cognata build their class definitions.
Comment 11: Lines 330-353: you describe some work which is closely related to yours which is fine and necessary, but it is much more useful to use a section such as this to also inform the reader about the weaknesses in the prior work of others that you will be addressing in the work following in this paper.
Response 11: This is an excellent suggestion. The "Closely related research" section (now Section 3) has been significantly reworked. It now critically analyzes the limitations of prior work, particularly how conventional two-category (ROI/background) methods often fail to substantially improve the legibility of small but vital elements like traffic signs. This revised section now better highlights the gap in the literature and motivates the novel contribution of our multi-category framework.
Comment 12: Figure 2: The meaning conveyed by (c) and (d) is very unclear. You need to explain what is being illustrated more clearly.
Response 12: Thank you for pointing this out. The text descriptions accompanying Figure 2 have been expanded for clarity. We now explicitly state that Figure 2(c) illustrates the grouping of the semantic map's pixel classes into our three proposed categories (background, weak ROI, and strong ROI). We further clarify that Figure 2(d) shows the result of the next step, where these pixel-level categories are converted into CTU-level assignments based on a pixel count threshold.
Comment 13: Section 5 (and possible elsewhere): You need to be very careful in section 5... to make sure that when you are referring to a single compression method with multiple categories, that you decide on a term and stick with it... Otherwise, there is room for reader confusion.
Response 13: Thank you for identifying this inconsistency. The manuscript has been carefully edited to ensure consistent terminology. We now consistently use hyphenated terms such as "three-category ROI compression" and "two-category ROI compression" to avoid any ambiguity.
Comment 14: Line 599-560: Table 2 seems to be strangely placed on the page for the version I have. Looking back, I notice that this is the case also in some earlier cases, (e.g. Fig. 1, Table 1 and others).
Response 14: We apologize for the formatting errors. We misunderstood the template. The placement of all figures and tables has been corrected throughout the manuscript to ensure they are properly aligned within the text columns.
Comment 15: Line 658-661: You should note that this is achieved at a cost to quality in the background category, which you believe represents a worthwhile improvement overall … or similar such text.
Response 15: Thank you for this suggestion to improve the conclusion. The text has been revised to explicitly state that the significant improvement in critical ROI areas is achieved by reducing the fidelity of less critical background elements, and that this trade-off results in a net improvement for the remote driving task.
Response to Comments on the Quality of English Language
We thank the reviewer for spotting these errors and helping us improve the manuscript's readability. All suggested changes have been made.
- Point 1 (Line 13): The full name for the HEVC abbreviation has been added to the abstract.
- Point 2 (Line 13): A citation to our preliminary study [49] has been added in the "Closely related research works" section.
- Point 3 (Line 38): “unpolite” has been corrected to “impolite”.
- Point 4 (Line 54): The sentence has been rephrased to be gender-neutral.
- Point 5 (Line 119): The sentence has been rephrased to be gender-neutral.
- Point 6 (Line 139-141): This sentence was part of a section that has been significantly revised for clarity and accuracy based on other comments. The confusing text regarding a 1000:1 ratio has been removed and replaced with more pertinent information.
- Point 7 (Section 1.2): The paper has been reorganized as suggested. Application-oriented context now appears earlier in the Introduction to make the paper's focus clearer from the outset.
- Point 8 (Line 213): "Coding Tree Unit (CTU)" is now written in full at its first use in the main text, in addition to the abstract and the list of abbreviations.
- Point 9 (Line 241): A proper citation for the work of Li et al. [30] has been added.
- Point 10 (Line 244): A citation for Kvazaar [32] has been added at its first mention.
- Point 11 (Line 251): The typo has been corrected from “compared” to “compare”.
- Point 12 (Line 251): The reference number for Li et al. [30] has been added.
- Point 13 (Line 254-255): The broken reference has been corrected.
- Point 14 (Figure 2): The lettering in Figure 2 has been edited to improve clarity.
- Point 15 (Line 450, etc.): "describe" has been corrected to "describes" in all indicated instances.
- Point 16 (Line 584): The missing table number (Table 2) has been added.
Reviewer 2 Report
Comments and Suggestions for Authors
The article describes an interesting application of ROI video coding to remote operation of vehicles. The tests is sound and the results are consistent and believable.I would recommend an evaluation of the ROI encoded video against the standard encoded video made by actual human remote vehicle operators, to measure the effectiveness of the video encoding strategy (i.e. a kind of task based video quality evaluation). PSNR and SSIM, especially in the ranges listed in the results can be misleading as quality indicators. Another important quality indicator that is not measured is the temporal variation of the ROIs quality that may be induced by variations both in the shape of the ROIs and variations of the QPs used for each ROI along time. Temporal variations of quality can have a very detrimental effect in visual perception.
Comments on the Quality of English LanguageThe text has a few typos and bad uses of English that should be corrected.
Author Response
We sincerely thank the reviewer for their insightful feedback and constructive suggestions. The points raised are valuable, and we have addressed them in our revisions.
Response to Reviewer Comments
Comment 1: Human-Based Evaluation The reviewer notes that while the tests are sound, an evaluation of the ROI-encoded video by actual human operators would be beneficial to measure the true effectiveness of the strategy, as PSNR and SSIM can be misleading.
Response: Thank you for this excellent suggestion. We completely agree that a task-based evaluation involving human remote operators would be the ultimate measure of our method's practical effectiveness. We also acknowledge the limitations of using objective metrics like PSNR and SSIM alone for assessing perceptual quality.
The primary focus of this paper was to establish a novel multi-category compression framework and validate its objective performance in reallocating bandwidth to enhance critical regions. A formal human factors study represents a significant and important next step, but it is beyond the scope of the current work.
We have added this point to the "Limitations" and "Future Work" sections of our manuscript to explicitly state that task-based human evaluation is a critical area for subsequent research.
Comment 2: Temporal Quality Variation The reviewer raises the important point that the paper does not measure the temporal variation of the ROI's quality, which can be detrimental to visual perception.
Response: This is a very insightful point, and we thank the reviewer for bringing it to our attention. We agree that temporal inconsistencies, such as flickering caused by rapid changes in ROI shapes, coverage, or QP values, can negatively impact the operator's experience.
Analysing these temporal dynamics is a complex but important topic. While a full analysis is outside the scope of this paper, we recognize its importance. We have acknowledged this as a limitation and a key direction for future work in the revised manuscript. Future research should investigate methods to ensure temporal stability in quality across the different ROI categories.
Response to Comments on the Quality of English Language
Comment: The text has a few typos and bad uses of English that should be corrected.
Response: Thank you for noticing these issues. The manuscript has been thoroughly proofread and edited to correct typos and improve the overall quality of the English language for better clarity and readability.
Round 2
Reviewer 1 Report
Comments and Suggestions for Authors
The alterations made to the original submission greatly improve clarity in the purpose of the work described and the original contributions it contains, including its context and limitations.
I am now completely satisfied with the quality of the paper in all respects.